# Using Objective Bayesian Methods to Determine the Optimal Degree of Curvature within the Loss Landscape

## Abstract

The efficacy of the width of the basin of attraction surrounding a minimum in parameter space as an indicator for the generalizability of a model parametrization is a point of contention surrounding the training of artificial neural networks, with the dominant view being that wider areas in the landscape reflect better generalizability by the trained model. In this work, however, we aim to show that this is only true for a noiseless system and in general the trend of the model towards wide areas in the landscape reflect the propensity of the model to overfit the training data. Utilizing the objective Bayesian (Jeffreys) prior we instead propose a different determinant of the optimal width within the parameter landscape determined solely by the curvature of the landscape. In doing so we utilize the decomposition of the landscape into the dimensions of principal curvature and find the first principal curvature dimension of the parameter space to be independent of noise within the training data.

## 1 Introduction

When training a neural network we aim to find a parametrization which minimizes the variance of the data around the model's conditional mean value. A statistic which is reflective of this variance is known as a loss function and can be seen as creating a landscape mapping a model parametrization to a corresponding loss value. Thus, higher points in the landscape reflect higher loss values and a worse model parametrization. The saliency of other features of the loss landscape on the model performance are relatively less clear and in some cases are points of contention within the field. One such point is whether the width of a basin in the landscape surrounding a local minimum (we will also refer to this as the width of the minimum) is reflective of the ability of a model parametrization at the minimum to generalize to unseen data. It is a common notion that the wider the minimum in the landscape, as measured by the Hessian matrix of the loss function (Keskar et al., 2016; Dinh et al., 2017), the better the model parametrization will generalize. The intuition behind such a belief is simply that, wider minima reflect that a model will experience less deviation in its loss metric as a result of minor deviations of its parameter values. As a result the model is more robust than if it were to be parametrized by a very specific parameter set found at a sharp minimum.

In this work we aim to demonstrate that the width of a minimum is a key feature of the loss landscape and provides significant information on the progress of the training of a model. We deviate, however, from the views of the field that the widest minima provide the best generalizability by reflecting that there is instead an optimal width or curvature around the parametrization with the best generalizability which is not necessarily the widest point in the landscape. To this end Section 2 provides the necessary background information that we will utilize in developing our theories which are presented in Section 3. Section 4 and Section 5 then provide empirical evidence in support of the theoretical findings with Section 4 describing the methods employed to test the theories. Section 5 then provides and discusses the results of these empirical tests. Finally we conclude in Section 6 with our closing remarks.

The contributions of this work are threefold. Firstly we evaluate the concept of Energy-Entropy competition of neural networks (Zhang et al., 2018) in the context of the bias-variance trade-off (Geman et al., 1992) and reflect that a correlation exists between energy and entropy as opposed to

a competition or trade-off as was first presented. Secondly we utilize the Fisher Information of the loss landscape in the area of a minimum to reflect that an optimal level of curvature exists within the landscape which does not necessarily occur at the point in the landscape with the least curvature. Further to this, we provide a novel view on the overfitting of models to their training data using the loss landscape. Finally, the Fisher Information is utilized in defining the objective Bayesian prior known as the Jeffreys prior and we show that the test error of the model reaches its minimum value at the point in the landscape which corresponds to the Jeffreys prior. In addition, we show that at this point in the landscape the dimension of principal curvature of the model is at its maximum entropy. In doing so we also reflect the noise invariance of the dimension of principal curvature.

## 2 BACKGROUND

### 2.1 FISHER INFORMATION AND UNINFORMATIVE (JEFFREYS) PRIORS

The Fisher Information (which we denote by $\zeta(\theta)$) is a metric dependent on the model parametrization which measures the amount of information that a sufficient statistic based on the observable data, such as the variance of the data around the model predictions (Jaynes, 2003), carries about an unknown parameter $\theta$. In the case of a Gaussian model, the Fisher Information is equal to the Expected Hessian of the log-likelihood of the Gaussian. The necessary regulatory conditions for this equality to be true apply to the entire exponential family of distributions, however, in our case it is sufficient for this to hold for the Gaussian distribution (Ly et al., 2017). One of the key properties of the Fisher Information Matrix is that its determinant is invariant under reparametrizations of a trained model. Thus, when the parameters used in modelling a distribution are changed, the Fisher Information in each dimension will change, however, the determinant or volume of information remains unchanged between the model parametrizations.

The invariance property of the Fisher Information was the reason for its utilization in Jeffreys (1946) who sought to create a Bayesian prior with such an invariance property. The resultant prior is known as the Jeffreys prior and is shown in Equation 1, where $H(\theta)$ denotes both the Hessian and Expected Hessian matrices (we will treat the Expected Hessian and Hessian interchangeably for the remainder of this work, as is common in the literature (Zhang et al., 2018; Karakida et al., 2019)).

$$P(\theta) \propto \sqrt{\det \zeta(\theta)} = \sqrt{\det H(\theta)} \qquad (1)$$

As has been shown in Jaynes (1968; 2003) the utility of the Jeffreys prior is not limited to the invariance property, as the Jeffreys prior is an example of an uninformative or objective prior, and as a result informs the posterior distribution as little as possible. The Jeffreys prior is thus used to reflect complete prior ignorance about the correct model parametrization, resulting in a posterior distribution with parameters completely determined by the observed data. A key perspective of this property is that the Jeffreys prior, thus, imposes a uniform distribution over the function space of the model, not the parameter space. This is due to the density of the prior being inversely proportional to the Hessian at a certain parametrization, and as such places higher density on parametrizations with a unique function approximation and low parameter variance. This would result in an even distribution over the function approximations and as a result the choice between function approximations is left to the model learning from the data. The relationship between the Hessian and the parameter variance of a model is discussed further in Section 2.2 below.

### 2.2 THE BIAS-VARIANCE DILEMMA, ENERGY-ENTROPY COMPETITION AND MINIMUM DESCRIPTION LENGTH

It is a well-known fact that a model learning to equate its conditional mean precisely to the values found in the training data is not always beneficial to the performance of the model on unseen data. In particular when we observe a decrease in training error but increase in validation or test error we say that the model is overfitting the training data (Hawkins, 2004). In Geman et al. (1992) it is shown empirically that to decrease the variance of the data around a model's predictions (reduce the training error) it is necessary for the variance in the model parameters to increase. Further, Geman et al. (1992) reflect that a large parameter variance corresponds to the overfitting of the model to the training data. This trade-off between the bias of the model and the variance of its parameters is

known as the Bias-Variance Dilemma (Sammut & Webb, 2011). In Geman et al. (1992) the only means presented to mitigate this dilemma is to obtain more training data.

We see, however, that neural networks are capable of learning complex tasks with limited data and even generalize in spite of the Bias-Variance Dilemma. In Zhang et al. (2018) it is argued that the success of neural networks is due to a bias of Gradient Descent towards wider minima in the loss landscape. To reflect this, Zhang et al. (2018) derive a Bayesian posterior distribution for the parameters of a model given the training data. To derive this distribution, Zhang et al. (2018) utilize a Gaussian likelihood with conjugate Gaussian prior, which we generalize in Equation 2 by allowing any prior distribution $exp(h(\theta))$ which results in a proper posterior distribution. The exponential term of this generalized prior $h(\theta)$ is seen as some function of $\theta$ while $f(x_i, \theta)$ denotes the function approximation by the neural network. $\sigma_i^2$ is the variance of the output corresponding to data point $x_i$, and finally $y_i$ is the true output for a particular $x_i$ in the training data. The derivation of Equation 2 can be found in Appendix A.1.

$$P(\theta|\mathbf{X}) = \frac{1}{Z} \exp\left[-\left(\sum_{i=1}^{P} \frac{(y_i - f(x_i, \theta))^2}{2\sigma_i^2} - h(\theta) + \frac{1}{2} \log det(H(\theta))\right)\right] \qquad (2)$$

From Equation 2 we see that to maximize the probability of a parametrization we must simultaneously maximize the model likelihood (by minimizing the first term in the exponential), the prior probability of the parametrization and the model entropy. The model entropy is reflected by the final term in the exponential and is inversely proportional to the Hessian of the loss landscape at the parametrization. Using the posterior distribution Zhang et al. (2018) note that maximizing the model likelihood is not the only factor which should be used in determining the model parametrization, and in some cases it may be beneficial to trade-off some training error for an increase in model entropy, which the authors called Energy-Entropy competition. Zhang et al. (2018) state that the bias of Gradient Descent towards wider minima, with smaller Hessian values, results in the model naturally maximizing entropy, aiding in its generalizability. We see, however, from the Bias-Variance Dilemma that by reducing the bias of the model on the training data, and increasing its likelihood, that the model entropy will naturally also increase due to the higher variance in the parameter values.

With the perspective of both the Bias-Variance Dilemma and Energy-Entropy competition we see that wide points in the loss landscape have been related to both overfitting and improved generalizability of a model parametrization. Thus, from one perspective we aim for sharp minima within the landscape and from the other we should aim for wide minima. The issue of the width of a minimum is further confounded by Dinh et al. (2017) which states that the width of a minimum is not a consistent indicator of the ability of a model to generalize. The impact of the width of a minimum in the landscape is still an open question and one which we try address in this work.

The Minimum Description Length (MDL) Principle is an information theoretic principle which states that the optimal model for a set of data provides the best compression of the data (Rissanen, 1978). In other words the optimal model is the simplest model which incurs the least training error. This principle is another example of the trade-off between model complexity and minimizing the model bias. Due to its assertion that the optimal model is the simplest unbiased one the MDL Principle is the mathematical formulation of Occam's Razor, and is expressed by Equation 3, which reflects that to compress the data $D$ optimally we must find the parametrization with the net minimum entropy in the parameter space $L(\theta)$ and in the data given the parametrization $L(D|\theta)$.

$$L(D) = \min_{\theta \in \vartheta}(L(\theta) + L(D|\theta)) \qquad (3)$$

For the exponential family of likelihood distributions the Jeffreys prior is used to enforce the MDL property on the posterior distribution and results in a minimax optimal posterior, which is to say that the maximal risk of the model parametrization is minimal out of all unbiased parametrizations (Lehmann & Casella, 2006). The minimax optimality property, thus, provides the lowest upper bound of the risk for all model parametrizations. Thus the MDL property is related to the Bias-Variance Dilemma and MDL posterior distributions aim to avoid overfitting.

A final necessary principle which encompasses all of the topics above is the Likelihood Principle (Jaynes, 2003), which states that within the context of a specified model, the likelihood distribu-

tion $L(D|\theta)$ trained from data $D$ contains all the information about the model parameters $\theta$ that is contained in $D$.

## 2.3 PRINCIPAL CURVATURE

The Jeffreys prior, and by extension the Fisher Information, finds further utility in its use as a right Haar measure for the parameter space of a normal distribution (Berger, 2013). The Haar measure is used to assign an invariant volume to a subset of locally compact topological groups and, thus, forms an integral over the compact groups. In the case of the parameter space of the normal distribution the topological groups will be of parametrizations with similar function approximations and, thus, similar loss metric within the basin surrounding a local minimum in the loss landscape. Further, we note that the parameter space of a probabilistic model forms a statistical manifold and by extension a Riemannian manifold (Rao, 1945). The metric tensor for statistical manifolds is the Fisher Information metric (Skovgaard, 1984), defined as the expected value of the individual elements of the Fisher Information matrix, which forms the tangent space of such manifolds. As stated in Section 2.1, in the case of Gaussian parameter spaces the Fisher Information Matrix can be equally derived as the Hessian matrix of the loss function relative to the model parameters. This is significant as the Hessian matrix is used in the area of a critical point on a Riemannian manifold for obtaining the shape operator (Spivak, 1970), and as a result the principal curvatures at the point (Porteous, 2001). In the case of a Gaussian parameter space the shape operator is the Gaussian curvature defined as the determinant of the Hessian matrix $det(H(\theta))$ (Koenderink & Van Doorn, 1992). The principal curvatures are defined as the eigenvalues of the Hessian matrix and decompose the manifold into orthogonal dimensions of curvature, with the first eigenvector reflecting the dimension of most curvature.

It is important to note that while the parameter space of a statistical model forms a Riemannian manifold, when parametrized by an overly-determined model such as a neural network, the parameter space will not be Riemannian but rather semi-Riemannian, due to the fact that the Fisher Information metric will no longer be defined over the entire manifold. Such undefined points for the metric are a result of a singular metric tensor at the model parametrization and occur due to the covariance of parameters within the model. Covariant parameters necessarily occur with the addition of hidden layers to the model and result in dimensions on the manifold in which the parameters may be varied without altering the behaviour of the model. This results in dimensions of no curvature along the manifold. As seen in Section 5, this is not a destructive point for the training procedure, however, we must necessarily remain cognisant of such covariant dimensions along the statistical manifold.

## 3 MODEL ENTROPY, THE LOSS LANDSCAPE AND GENERALIZATION

The aim of training a neural network is to find the most probable parametrization for a model as determined by the posterior probability reflected by Equation 2. This is achieved by maximizing the combination of the likelihood, prior probability and entropy of the model parametrization. The likelihood we increase normally by decreasing the variance of the training data around the model predictions. The entropy term we have no direct control over as the landscape is completely determined by the data, the sufficient statistic being used to determine the parameters (which is the loss metric) and the hypothesis (the model architecture being trained). So the only component of the posterior left to be determined is the prior. As with most work in Bayesian statistics this is the most difficult part and must be treated with great care. There are presently two common approaches to setting this prior distribution, the first of which being to not specify one, or more precisely use an implicit uniform prior (Chaudhari & Soatto, 2018) and, thus, use maximum likelihood estimation to determine the parameter values. The second common approach is to utilize a conjugate Gaussian with a mean of $0$ for the prior. In practice this method takes the form of L2 regularization, also known as weight decay (Krogh & Hertz, 1992), with $h(\theta) \propto ||\theta||^2$ in Equation 2. Neither approach has proven to be sufficient consistently for deterring models from overfitting, without introducing a form of bias, due to their unjustified assumptions about the correct model parametrization. This is relatively clear in the conjugate Gaussian prior approach which assumes a mean of $0$ for the parameter values. In a case where we have explicit prior knowledge that such a mean and distribution is in-fact correct for the model parameters then this would be a correct approach, however, in almost every case we are completely ignorant to the values of the true parametrization and, thus, we

would be biasing our models to some degree by using this prior. It is necessary when developing an unbiased model that this absence of prior knowledge be reflected in the training procedure.

The source of error from the uniform prior is slightly more nuanced, as initial intuition would suggest that giving equal probability to all values a parameter could take is a correct means of reflecting our prior ignorance about the parameter values. However, this method fails to accurately reflect the probabilities that unlikely parametrizations may be correct and in a sense exhibits a kind of confirmation bias. As discussed in Section 2.2 the model which is optimal is the one which has the lowest variance of the data around its predictions $L(D|\theta)$ while maintaining low variance in the model parameter space $L(\theta)$. In addition the bias-variance trade-off (Geman et al., 1992) state that to decrease data variance we must increase parameter variance reflecting the trade-off between $L(\theta)$ and $L(D|\theta)$ in the MDL equation. Further, we related this to the Energy-Entropy competition concept (Zhang et al., 2018), where it was also stated that neural networks are biased towards wide minima. Hence, by placing equal prior probability on all areas of the landscape we see that the use of the uniform prior will result in a posterior distribution over the model parameters which places excessive density on high variance areas of the landscape while at the same time places too little probability on very specific, low variance parametrizations of the model. This results in the development of a sub-optimal model which favours wider minima within the loss landscape and as a result excessively reduces the variance of the data around its predicted values.

Thus we conjecture that a correct prior for a model would be the Jeffreys prior shown in Equation 1 and Equation 4:

$$P(\theta) \propto \sqrt{\det \zeta(\theta)} = \sqrt{\det H(\theta)} \tag{4}$$

As a result we see in Equation 2 that such a prior would give the equation for $h(\theta)$ as

$$h(\theta) = \frac{1}{2} \log det(H(\theta)) \tag{5}$$

We note that, with the use of the Jeffreys prior, the prior and entropy term in the posterior formulation cancel out, leaving the likelihood term as the only factor determining the posterior probability, as can be see in Equation 6.

$$
\begin{aligned}
P(\theta|\mathbf{X}) &= \frac{1}{Z} \exp\left[-\left(\sum_{i=1}^{P} \frac{(y_i - f(x_i, \theta))^2}{2\sigma_i^2} - \frac{1}{2}\log det(H(\theta)) + \frac{1}{2}\log det(H(\theta))\right)\right] \\
&= \frac{1}{Z} \exp\left(-\sum_{i=1}^{P} \frac{(y_i - f(x_i, \theta))^2}{2\sigma_i^2}\right)
\end{aligned}
\tag{6}
$$

It is, thus, possible to utilize the loss landscape in the area of a minimum to determine the degree of certainty we may have in our model parametrization being the true parametrization and as a result determine the necessary Jeffreys prior probability. This is due to the fact that higher entropy means wider minima which reflects higher parameter variance and the necessity to be less certain of the parametrization in that area. The opposite is true for an increase in certainty in our parametrization at a sharp minimum. Furthermore, this would mean we are objectively setting our prior based on the model behaviour given the observed data and sufficient statistic. Note, we do not say we determine our prior based on the hypothesis as the determinant of the Fisher Information/Hessian is invariant under reparametrizations. This means that in the area of a minimum, by transforming the hypothesis to be modelled by an alternate set of parameters $\theta'$ the dimensions and volume of the landscape will adjust such that $\sqrt{det(H(\theta))} = \sqrt{det(H(\theta'))}$ (Fisher, 1922). A necessary distinction regarding the Jeffreys prior is that, while it places the full parameter determination on the data, it does not necessarily result in a posterior distribution which has extracted all information from the data. Information which provides an insufficient decrease in data variance to warrant the increase in variance in the model parameters will not be utilized as the model naturally "distrusts" this information by providing a relatively lower prior probability to the more entropic parametrization found in the wider basin. This is where we see the utility of the Jeffreys prior with regard to the MDL property reflected by Equation 3 as it balances both model complexity $L(\theta)$ while fitting the data $L(D|\theta)$.

The primary power of the Jeffreys prior comes from the use of the Fisher Information. Naturally as the model fits the data and captures information, the amount of information left in the data which

remains uncaptured by the model decreases. This is observed as a decrease in the Fisher Information. We see, however, that the model entropy increases as the Hessian matrix, and by extension the Fisher Information, decreases, which is again in agreement with the bias-variance trade-off. The consequence of this observation is that to capture all the information from a sufficient statistic determined by the training data we must utilize increasingly complex models, capable of modelling finer details found in the data. The utility of such fine details to the model performance exhibits diminishing returns to a point where the perturbation of a parameter capturing these details will not result in any significant deviation in the model behaviour. Simply, as more information is shared between parameters, the individual importance of a parameter decreases. This is in contrast to an under-parametrized model where the parameters capture as much of the most important information from the sufficient statistic as possible and rely heavily on this information in determining its behaviour. In light of the Fisher Information we see the Maximum Likelihood Principle further reflects that the use of the uniform prior biases our models towards maximum entropy within the loss landscape by extracting all information from the training data at the expense of higher model complexity.

It must be noted that the propensity of neural networks to extract all information from the training data is not an inherently negative quality of the models. Quite the opposite, it is reflective of the power and capability of the models which are designed to learn the variances within data and utilize this information in determining their behaviour. As a result, however, the efficacy of these models is directly related to the reliability of the data on which they are trained and for all the information found in the training data to be present and reflective of the entire population of data being modelled. This is seldom the case as training data is inevitably noisy, either due to noise from sampling and capturing of the data, or due to confounding aspects of the task domain which on average do not affect the population data distribution but do provide a source of structured noise when their effect is observed on the training data. Minimizing the Fisher Information metric and fulfilling the Maximum Likelihood Principle in such cases would reflect that the information found in the noise of the training data was utilized in determining the model parameter values, which is clearly undesirable and is known as the model over-fitting the training data (Hawkins, 2004).

We, thus, see that the notion of the widest possible minima in the loss landscape providing the best generalization performance is only true in a noiseless environment. The view, however, that wide areas in the landscape generalize better is true as this width in the landscape would merely reflect that the model has captured more information from the data than a parametrization found in a steeper portion of the landscape. Naturally this would provide better test error performance by the model if it has captured the information found in the training data which is reflective of the information within the population data distribution being modelled. We conclude that the width of the landscape in which a model finds itself is demonstrably important and that there is a precise width in the landscape which provides the model with the best possible test error performance. This point would be exactly where the model prior is equal to the Jeffreys prior as determined by the Fisher Information of the loss landscape. This is due to the aversion of the Jeffreys prior to any information which does not justify the increase in model entropy by a superior decrease in data variance or prediction error, while remaining objective in the sense that the prior is completely determined by the loss landscape and has as little effect on the parameter posterior distribution as possible. Thus, noise in the training data which only serves to perturb and hinder the learning of the model without providing sufficient benefit to how the training data is fit will not be learned by the model.

## 4    METHODS

From the above discussion it is clear that it is not enough to merely reduce the variance of the data around the model prediction as it is possible for the model to reduce this variance to an excess degree. We, thus, require a metric for the difference between the model and true distributions which is only minimized when the two distributions are identical. This is not the KL-divergence as this metric merely reflects the density of a distribution $B$ which lies outside of another distribution $A$. It is, thus, possible to minimize the KL-divergence while the distributions are not identical by having distribution $A$ surround or encapsulate distribution $B$. We will, thus, use the Jeffreys divergence as the difference metric between the two distributions, as the Jeffreys divergence is uniquely minimized when the two distributions are identical. The Jeffreys divergence is shown below in Equation 7 and is merely the sum of the KL-divergence for the true parameter distribution $T(\theta)$ compared to the

model parameter distribution $P(\theta|\mathbf{X})$ and the opposite KL-divergence. We show the derivation of the Jeffreys Divergence from this summation in Section A.2 of the Appendix.

$$D_J(T(\theta)||P(\theta|\mathbf{X})) = \int (T(\theta) - P(\theta|\mathbf{X}))(lnT(\theta) - lnP(\theta|\mathbf{X}))d\theta \tag{7}$$

Naturally Equation 7 is intractable due to the necessity to integrate over all parametrizations. However, as discussed in Section 3, the use of maximum likelihood estimation results in an excess of density on high variance parametrizations in the posterior parameter distribution in Equation 2. It was thus sufficient to evaluate the Jeffreys Divergence at a single point in parameter space and observe the relative densities at that point, providing a distance metric as opposed to a divergence metric. We will, thus, refer to the Jeffreys Distance for the remainder of this work, with the formula shown in Equation 8.

$$D_J(T(\theta)||P(\theta|\mathbf{X})) = (T(\theta) - P(\theta|\mathbf{X}))(lnT(\theta) - lnP(\theta|\mathbf{X})) \tag{8}$$

The necessity of a distance metric being positive semi-definite is upheld by this metric as it is clear when $(T(\theta) - P(\theta|\mathbf{X})) < 0$ then $(lnT(\theta) - lnP(\theta|\mathbf{X})) < 0$. Likewise when $(T(\theta) - P(\theta|\mathbf{X})) > 0$ then $(lnT(\theta) - lnP(\theta|\mathbf{X})) > 0$. This is a benefit of the symmetrical property of the Jeffreys Divergence which the KL-divergence does not possess. Thus, substituting the model posterior formula from Equation 2 as well as the true model distribution in Equation 9 into the logarithmic terms of Equation 8 we obtain the formulation shown in Equation 10.

$$T(\theta^*) = \frac{1}{Z^*} \exp\left(-\sum_{i=1}^{P} \frac{(y_i - f(x_i, \theta^*))^2}{2\sigma_i^2}\right) \tag{9}$$

$$(lnT(\theta) - lnP(\theta|\mathbf{X})) = \left(-\sum_{i=1}^{P} \frac{(y_i - f(x_i, \theta^*))^2}{2\sigma_i^2} + \sum_{i=1}^{P} \frac{(y_i - f(x_i, \theta))^2}{2\sigma_i^2}\right.$$
$$\left. -h(\theta) + \frac{1}{2}\log det(H(\theta)) + Z^* - Z\right) \tag{10}$$

Assuming now that $\theta = \theta^*$, and thus $f(x_i, \theta) = f(x_i, \theta^*)$, as would be the case at the end of an unbiased training procedure, we see that the two variance terms in Equation 10 will cancel out. Further, we see that the only means of obtaining a 0 value for the expression is to use the Jeffreys prior causing $h(\theta)$ to cancel with the entropy term $\frac{1}{2}\log det(H(\theta))$, as discussed above in Section 3, with Equation 5 and Equation 6. Finally we see that using the Jeffreys prior would result in the posterior model distribution shown in the last line of Equation 6. If $f(x_i, \theta) = f(x_i, \theta^*)$ then it is clear that $Z = Z^*$ is the necessary corresponding normalizing constant, and, thus, these terms will also cancel out in Equation 10. A similar argument can be made for the probabilities component of the Jeffreys distance $(T(\theta) - P(\theta|\mathbf{X}))$ in Equation 8, whereby we equate the two likelihood terms, then the use of the Jeffreys prior ensures that the posterior distributions do not differ resulting in a Jeffreys Distance of 0.

From Equation 10 we see that the use of the Jeffreys prior while minimizing the error of the model directs the model to a parametrization which results in the model likelihood being equal to that of the true underlying distribution likelihood. This allowed us to determine the point at which the model found the most probable parametrization in the landscape corresponding to the use of the Jeffreys prior, as being the parametrization which equated the model likelihood and true distribution likelihood values. In light of the bias-variance trade-off this would also mean that if the model reduces its bias to a greater extent that it would have moved to a point of excess parameter variance and model entropy as well of an excessively low Fisher Information metric. As expressed in Section 3 this is indicative of the use of noise by the model in determining its parameter values. As a result the model would have overfit the training data. We, thus, reach the intuitive conclusion that once a model parametrization becomes a more likely distribution to have generated the training data than the underlying true distribution itself, that the model will have began to overfit the training data.

The first empirical result presented in Section 5 is, thus, a distribution reflecting the proximity of the training step at which minimum test error was reached to the training step at which the model likelihood is equal to that of the true underlying distribution likelihood (we will refer to this as the likelihoods intersecting) over a number of network training procedures. The aim of this experiment is to empirically confirm that the Jeffrey Prior parametrization provides the minimum test error for a model. For this experiment it is necessary to possess a ground truth on the likelihood of the true data generating distribution on the training data. Unfortunately such ground truth information is not available on real-world data sets. As a result it was necessary for our experiments to use synthetic data which was generated by a ground truth network, which we shall refer to as the "True" network. A "Training" network will then learn to model this ground truth network on noisy training data.

Hence, the procedure for this first experiment is as follows. We create a randomly generated True network with depth between 5 and 15 layers. The widths of the model layers are randomly sampled between 5 and 100 neurons. The layers are sorted in descending order of width, as is common for network architectures used in practice and results in the wider, earlier layers extracting features for the later layers. We then prepend the 100 neuron input layer and append the 1 neuron output layer. All layers except the last are sigmoidal. This is the model used to generate data. We then randomly initialize our training network. The number of layers in this network is randomly chosen from the range of $[TrueNetworkSize + 5, 25]$ to ensure we obtain a sufficiently large network to overfit the data. The widths of this network's layers are sampled from the range of $[TrueNetworksSmallestLayer, 100]$. This is again to ensure the model is over-parametrized. The True networks parameter values as well as the Training networks initial values are sampled uniformly from $[-1.0, 1.0]$ with a random positional bias added to the True network parameters in the range of $[-0.5, 0.5]$. This bias is to ensure the Training network starts with a significant degree of error. Finally, we utilize randomly sampled values between $[0.0, 1.0]$ as input to the models, with a training batch size of 50 data points and a test batch size of 500. This data is input to the True network and we obtain the corresponding data labels as output. Lastly we add Gaussian noise to the Training data only (while the Test data remains clean) with a mean of 0 and variance of 0.2. The Training network is then trained to model the True network using this data and we observe the points where the likelihoods intersect and where the test error is minimized. This process was repeated for 935 separate training procedures. The distribution of the distances between the likelihoods intersecting and the minimum test error are shown below in Section 5.

Having observed the relationship between test error and the relative likelihoods of the training and true networks, we then decompose the parameter landscape into its dimensions of principal curvature to reflect the impact of noise in the training data on the landscape by first training on noiseless data until a certain training error is achieved (error $< 0.4$), at which point the noise is added to the data. We observe and interpret the resulting impact of the addition of the noise on the curvature of the landscape. Secondly we aim to observe the relationship between the dimensions of principal curvature and the generalizability of a model parametrization by observing the entropy on a Riemannian sub-manifold of the original statistical manifold (Chen, 2019), defined over the primary dimensions of principal curvature, relative to the test error and likelihood values of a model parametrization.

Due to the necessity to calculate a full Hessian matrix [1] for these experiments a smaller network was utilized, than in the first experimental procedure. The procedure is the same as described above for the first experimental procedure, however, in this case the training model was composed of 4 hidden layers with widths of 25, 20, 12 and 7 neurons respectively. As shown empirically in Karakida et al. (2019), however, the behaviour of the Eigen-decomposition of the Fisher Information is independent of the size of the network architecture, and, hence, this smaller network is sufficient for observing the effect of noise on the dimensions of principle curvature of the model. Hence, the Hessian matrix calculated has a dimensionality of $831 \times 831$ elements. The training data was again obtained from a true network. In this case different permutations of 25-bit binary strings were input to this model which returned the corresponding scalar using one hidden layer of 5 neurons and a linear output layer.

---

[1] The Hessian matrix was calculated using the Jax library (Bradbury et al., 2018)

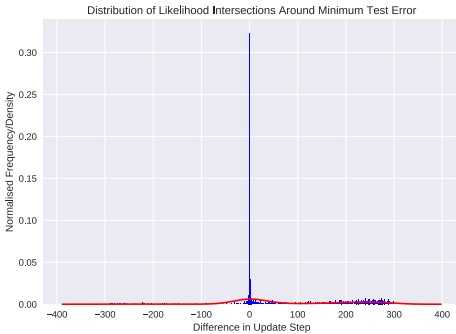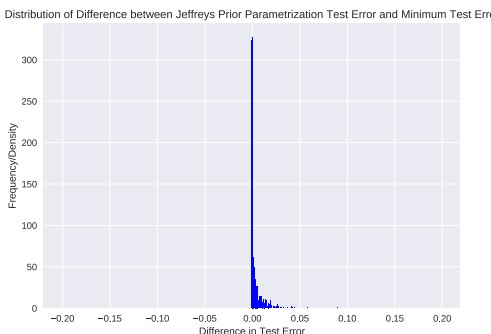

Figure 1: Distributions of the number of parameter update steps (left) and the difference in test error (right) between the Jeffreys prior parametrization and the minimum test error from 935 individual training procedures (Kernel Density Estimate shown in red on the left).

## 5 RESULTS

As stated in Section 4 the first empirical result aimed to determine if the point in the landscape which is most probable under the posterior distribution resulting from the use of the Jeffreys prior possesses the optimal test error performance. The results of this first experiment are presented in Figure 1, where the left image reflects the distribution of the number of parameter update steps between where the Jeffreys prior parametrization occurs (where we observe the intersection of the likelihoods) and where minimum test error occurs. We see in this image that the highest density is placed around 0, with a vast majority of training procedures having the Jeffreys prior parametrization coincide exactly with the point of minimum test error. This supports the assertion that the Jeffreys prior results in the parametrization with the best generalization performance. We do, however, observe a small uniform spread of density to the right of 0 in this image. We observed that this is due to the test error oscillating once the Jeffreys prior parametrization is reached. This is merely a result of our inability to fine-tune the learning rate for the individual training procedures of the randomly generated training networks, with significantly different architectures. To reflect the fact that the test error for the Jeffreys prior parametrization in the trainings where oscillations occur is negligibly different from the minimum test error we present the right image of Figure 1. In this image we plot the density of the difference in test error of the Jeffreys prior parametrization compared to the minimum test error of the 935 separate training procedures. To make the error independent of the size of the regression values being modelled we divide the error by the mean of the regression y-values (generally this value is around 2.0 for the respective training procedures). We observe that in all trainings the discrepancy in test error is less than 0.1, with almost all discrepancies being less than 0.05. These error discrepancies are negligible and, thus, these results empirically confirm our hypothesis that the Jeffreys prior parametrization corresponds to the minimum test error.

The results of injecting noise into the training data only once the model has been sufficiently trained on clean data can be see in Figure 2. In this figure we present the Eigenvalues of 3 of the 5 principal curvatures of the loss landscape. Thus, each value reflects the inverse of the variance of the model in the dimension of the corresponding Eigenvector and a lower value reflects a higher entropy in the given dimension. From these results we can see that the injection of noise results in a sudden increase in the entropy of the lower principal curvature dimensions but has no effect on the first dimension of principal curvature. As the Fisher Information is reflected by the entropy this would mean that the low principal curvature dimensions capture more information at the injection of the noise into the data, while the information captured by the first principal curvature dimension remains smooth throughout the training procedure. This would reflect a noise invariance of the principal curvature of the landscape and as a result we can see that the dimension of principal curvature in the landscape is exclusively responsible for the capturing of the true or primary data information. In light of this noise invariance we observed the entropy of a sub-manifold corresponding only to the dimensions of high curvature (Eigenvalues $> 1$) during the training of a model. The results of this experiment are shown in Figure 3. In agreement with Figure 1 we see that in the area of the Jeffreys prior parametrization (intersection of likelihood values), the test error reaches its minimum

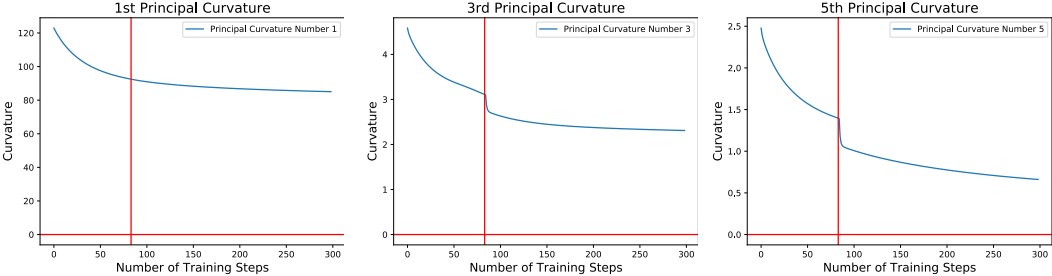

Figure 2: Impact of the addition of noise on 3 of the 5 Principal Curvatures of the Loss Landscape. Vertical lines signifies point of noise injection.

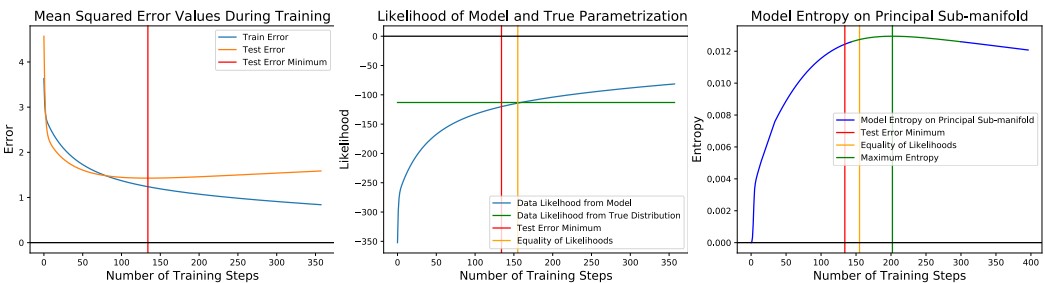

Figure 3: The Jeffreys prior parametrization is found in the area of parameter space minimizing test error and maximizing entropy on the principal sub-manifold.

value. A number of insights can be gained from the third image in Figure 3. Firstly, that the Fisher Information matrix and the Fisher Information Metric, are non-singular and positive semi-definite on this sub-manifold. This reflects that the dimensions responsible for capturing true information in the data are convex, with positive Gaussian curvature and that it is sufficient for the model to merely minimize this well-behaved region of parameter space. We, further, observe that the Jeffreys prior parametrization maximizes the entropy of this sub-manifold, reflecting that it still captures all true information from the data. We see the green portion of the entropy metric as being the area where the entropy is within $0.003$ of its maximum value. The fact that the entropy begins decreasing later in the training is reflective of the model forgetting true information while learning the noise once it starts overfitting. The fact that the entropy stagnates past the Jeffreys prior parametrization is due to the fact that the lower dimensions of principal curvatures in the sub-manifold were minorly sensitive to noise and that in this region the model is beginning to learn noise and forget true information at the same rate. We have, thus, demonstrated that maximizing entropy is beneficial in the absence of noise. However, when noise is present in the data, maximum entropy corresponds to overfitting.

## 6 CONCLUSION

We see that the notion of the width of the loss landscape being an indicator of a robust parametrization is correct, however, this is conditional upon the model being developed in a noiseless domain or, more significantly along a dimension of parameter space which is independent to the noise of the domain. With the aid of the Fisher Information perspective of the geometry of the landscape we see that the higher entropic points in the landscape directly reflect the absence of further information upon which the parameter values may be determined. Thus, we see the propensity of maximum likelihood models towards such high entropic points as a reflection of their propensity to utilize all information in determining the parameter values, including noise. Thus, we make the final conclusion that the point of maximum entropy in the loss landscape does not possess the best generalization performance and corresponds to the overfitting of the model to the training data. Instead the optimal point in the landscape occurs at maximum entropy in the dimension of principal curvature which corresponds to the most probable parametrization found by a Bayesian posterior distribution resulting from the use of the Jeffreys prior.

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

# A    APPENDIX

## A.1    DERIVATION OF THE NEURAL NETWORK BAYESIAN POSTERIOR DISTRIBUTION

The derivation presented below is adapted from Zhang et al. (2018).

We aim to derive a Bayesian posterior probability for the parameters of an artificial neural network conditional upon the training data. To this end we will assume a Gaussian likelihood distribution, as well as a general exponential prior. By this definition, this prior may be improper, however, without loss of generality we will assume that only priors which result in proper posterior distributions are utilized. Finally, we will assume that the data points are independent and identically distributed (i.i.d). Thus we utilize the following likelihood and prior distributions respectively:

$$P(\mathbf{X}|\theta) \propto \exp\left(-\sum_{i=1}^{P} \frac{(y_i - f(x_i, \theta))^2}{2\sigma_i^2}\right)$$

$$P(\theta) = \exp^{h(\theta)}$$

We, thus, obtain the posterior distribution, where $Z$ is the normalizing constant (also known as the partition function):

$$P(\theta|\mathbf{X}) = \frac{1}{Z} \exp\left(-\sum_{i=1}^{P} \frac{(y_i - f(x_i, \theta))^2}{2\sigma_i^2} + h(\theta)\right)$$

As stated in Zhang et al. (2018), machine learning problems where stochastic gradient descent has been successfully applied have an asymptotic data-to-parameter ratio of $P/N = O(1)$ as $P \to \infty$ where $P$ denotes the number of data points being used to determine $N$ parameters. This is referred to as the high dimensional limit (Advani et al., 2013). In this limit it is justified to perform Laplace Approximation:

$$\int P(\theta|\mathbf{X})d\theta \approx \frac{1}{Z} \sum_q \frac{\exp\left[-\sum_{i=1}^{P} \frac{(y_i - f(x_i, \theta_\mathbf{q}))^2}{2\sigma_i^2} + h(\theta)\right]}{\sqrt{\det H(\theta_q)}}$$

where $\theta_\mathbf{q}$ is a parametrization for a local minimum of a given loss function and $H(\theta_\mathbf{q})$ denotes the Hessian matrix of the model parameters at this minimum. We are, thus, summing over the various local minima within the loss landscape in the above formulation.

Finally, rewriting the denominator in an exponential form:

$$\frac{1}{\sqrt{\det H(\theta_q)}} = \exp\left(-\frac{1}{2}\log det(H(\theta_{\mathbf{q}}))\right)$$

We obtain:

$$\int P(\theta|\mathbf{X})d\theta \approx \frac{1}{Z}\sum_q \exp\left[-\left(\sum_{i=1}^{P}\frac{(y_i - f(x_i, \theta_{\mathbf{q}}))^2}{2\sigma_i^2} - h(\theta_{\mathbf{q}}) + \frac{1}{2}\log det(H(\theta_{\mathbf{q}}))\right)\right]$$

With the point density for the model parametrization at a single minimum in parameter space:

$$P(\theta|\mathbf{X}) \approx \frac{1}{Z}\exp\left[-\left(\sum_{i=1}^{P}\frac{(y_i - f(x_i, \theta_{\mathbf{q}}))^2}{2\sigma_i^2} - h(\theta_{\mathbf{q}}) + \frac{1}{2}\log det(H(\theta_{\mathbf{q}}))\right)\right]$$

## A.2 DERIVATION OF JEFFREYS DIVERGENCE

$$
\begin{aligned}
D_J(T(\theta)||P(\theta|\mathbf{X})) &= D_{KL}(T(\theta)||P(\theta|\mathbf{X})) + D_{KL}((P(\theta|\mathbf{X})||T(\theta)) \\
&= -\int T(\theta)ln\left[\frac{P(\theta|\mathbf{X})}{T(\theta)}\right]d\theta - \int P(\theta|\mathbf{X})ln\left[\frac{T(\theta)}{P(\theta|\mathbf{X})}\right]d\theta \\
&= -\int T(\theta)\left[lnP(\theta|\mathbf{X}) - lnT(\theta)\right]d\theta - \int P(\theta|\mathbf{X})\left[lnT(\theta) - lnP(\theta|\mathbf{X})\right]d\theta \\
&= \int -T(\theta)ln(P(\theta|\mathbf{X})) + T(\theta)ln(T(\theta)) - P(\theta|\mathbf{X})ln(T(\theta)) + P(\theta|\mathbf{X})ln(P(\theta|\mathbf{X}))d\theta \\
&= \int (-T(\theta) + P(\theta|\mathbf{X}))ln(P(\theta|\mathbf{X})) - (-T(\theta) + P(\theta|\mathbf{X}))ln(T(\theta))d\theta \\
&= \int (-T(\theta) + P(\theta|\mathbf{X}))(lnP(\theta|\mathbf{X}) - lnT(\theta))d\theta \\
&= \int (T(\theta) - P(\theta|\mathbf{X}))(lnT(\theta) - lnP(\theta|\mathbf{X}))d\theta
\end{aligned}
$$

