# OpenReview forum: "Using Objective Bayesian Methods to Determine the Optimal Degree of Curvature within the Loss Landscape"
_ICLR.cc/2020/Conference — Reject_

### Official Review · AnonReviewer1 · 2019-10-21
**Official Blind Review #1**

**Rating:** 1

**Review:**

This paper targets at a deep learning theory contribution based on information geometry. This contribution is tightly based on Zhang et al. (2018) and explains the generalization of deep learning from a Bayesian perspective. The main contribution the authors claimed is an optimal degree of curvature exist which gives the best generalization guarantees, which is in contrast to the commonly perceived "the wider the better".

First of all, the writing (including language etc) is of poor quality, to the extent that the submission is very difficult to read and can be rejected merely based on this, with unusual expressions, missing  punctuations, super long sentenses, and wongly used words. The reviewer won't list example here because they are everywhere.

What is even worse is the conceptral errors and defected derivations. For example, in eq.(1), the authors equate the Fisher information matrix (which is an expected Hessian) to the Hessian matrix, this is subject to conditions which must be clearly given right before/after the equation. As their results are largely based on the correctness of eq.(2), let's examine the derivations in appendix A.1.  In the first equation in A.1, what is the subindex "j"?  "Utilizing Laplace Approximation of the integral": such approximations have conditions that must be clearly stated. It is not clear how one can get the last approximation in page 12 from the previous equations. In summary, their eq.(2) is a loose approximation which is subject to a set of conditions (that are not given), and the derivation is of poor quality.

As a theoreiritical contirbution, the authors did not manage to converge to some simple and clear statements (theorems or equvalent). Instead, the contribution is largely *explanatory*. It is hard to observe anything new, given the poor writing and organization. The first 4 pages are mainly introductions of previous works.

The authors used information geometry and minimum description length to explain the generalization of deep learning. This is a small area. It is hard to miss closely related works by simple searching. Instead, the authors only cited Rissanen (1978). On the other hand, as the authors used the spectrum properties of the Fisher information matrix, there are some recent works by Amari which can be cited.

**Experience Assessment:**

I have published in this field for several years.

**Review Assessment: Checking Correctness Of Derivations And Theory:**

I carefully checked the derivations and theory.

**Review Assessment: Checking Correctness Of Experiments:**

I did not assess the experiments.

**Review Assessment: Thoroughness In Paper Reading:**

I read the paper at least twice and used my best judgement in assessing the paper.

---

> ### Author Response · Authors · 2019-11-13
> **Response to Official Blind Review #1 (Part 1 of 2)**
>
> We thank the reviewer for their constructive feedback.
>
> Q1) On the point of the poor writing quality.
>
> A1) We apologize for this and are working hard to improve the literary standard of the paper.
>
> Q2) In eq.(1), the authors equate the Fisher information matrix (which is an expected Hessian) to the Hessian matrix, this is subject to conditions which must be clearly given right before/after the equation.
>
> A2) Thank you for pointing this out. We will correct this error in the updated version.
>
> Q3) In the first equation in A.1, what is the subindex "j", "Utilizing Laplace Approximation of the integral": such approximations have conditions that must be clearly stated." and "It is not clear how one can get the last approximation in page 12 from the previous equations."
>
> A3) The merits of the Laplace Approximation are discussed in [1]. We are adapting the discussion from [1] for the updated version of Appendix A. We are in the process of improving the general quality of Appendix A, including the discussion on the assumptions of the derivation and making the link between certain steps in the derivation more explicit. Thank you for the feedback and assistance in improving this Appendix.
>
> Q4) As a theoreiritical contirbution, the authors did not manage to converge to some simple and clear statements (theorems or equvalent).
>
> A4) We believe Reviewer #2 provides an excellent summary statement, and one which we have included in the paper. Namely, "The authors provide theoretical arguments and claim that there exists an optimal width beyond which generalization can be poor".
>
> Q5)  It is hard to observe anything new, given the poor writing and organization.
>
> A5) We apologise if this was unclear and have made this clearer throughout the paper. In addition, we point to the last paragraph of the Introduction beginning at the bottom of Page 1 where we outline what we perceive to be our 3 main contributions. In summary:
> 1) We reflect that a correlation exists between energy and entropy as opposed to a competition or trade-off as was first presented in [1].
> 2) We reflect that an optimal level of curvature exists within the
> landscape which does not necessarily occur at the point in the landscape with the least curvature. We provide the novel perspective that the propensity of the model to find points of minimal curvature is a direct result of the model's propensity to overfit the training data.
> 3) We show that at the point in the landscape which corresponds to the Jeffreys prior the test error of the model reaches its minimum value and at this point the dimension of principal curvature of the model is at its maximum entropy. In doing so we also reflect the noise invariance of the dimension of principal curvature.
>
> Q6) The first 4 pages are mainly introductions of previous works.
>
> A6) We acknowledge that our work does rely heavily on past work and provides a detailed exposition of these past works, however, we view this as being necessary as we utilize a number of different field in this work. Namely Objective Bayes statistic, Information Theory, Differential Geometry and Machine Learning. We believe it necessary to not only provide sufficient background information for each field separately but also to illustrate the necessary overlap of the different concepts in these field for the full impact of this work to be seen. For example, a reader who is aware of the Jeffreys prior from an Objective Bayesian perspective may be unaware of its use as a right Haar measure in Differential Geometry. Thus, we aim to reflect the key fact that the Fisher Information, and as a result the Jeffreys Prior, is the commonality between the fields and guides our argument from the Objective Bayesian perspective of Section 3 to the Information Geometry perspective in Sections 4 and 5. We are, however, working at reducing the excess information in the paper, such as the overlap between the MDL property and Bias-Variance Dilemma, and restructuring aspects of our arguments to ensure that our contributions are clearer.
>
> Q7) The authors used information geometry and minimum description length to explain the generalization of deep learning. This is a small area. It is hard to miss closely related works by simple searching. Instead, the authors only cited Rissanen (1978).
>
> A7) Given the fact that the Minimum Description Length Principle can be equally phrased in light of the Bias-Variance trade-off which is also discussed in this work we see this as an opportunity to reduce the length of this work closer to 8 pages in line with the Conference standards and will, thus, rephrase our argument more in term of the Bias-Variance trade-off.

---

> > ### Author Response · Authors · 2019-11-13
> > **Response to Official Blind Review #1 (Part 2 of 2)**
> >
> > Q8) On the other hand, as the authors used the spectrum properties of the Fisher information matrix, there are some recent works by Amari which can be cited.
> >
> > A8) Based on your suggestion, we found the paper "Pathological spectra of the Fisher information metric and its variants in deep neural networks" by Karakida, Akaho and  Amari [2]. This is very exciting work which we will now include in this paper. Thank you for this helpful recommendation. Unfortunately this paper was only uploaded to ArXiv on the 14th October 2019 (after this conference's deadline) and as a result we could not include it in our original submission. None the less, this is a welcome opportunity to further contextualize our work in this paper.
> >
> > [1] Zhang, Yao, et al. "Energy–entropy competition and the effectiveness of stochastic gradient descent in machine learning." Molecular Physics 116.21-22 (2018): 3214-3223.
> > [2] Karakida, Ryo, Shotaro Akaho, and Shun-ichi Amari. "Pathological spectra of the Fisher information metric and its variants in deep neural networks." arXiv preprint arXiv:1910.05992 (2019).

---

### Official Review · AnonReviewer2 · 2019-10-22
**Official Blind Review #2**

**Rating:** 6

**Review:**

The paper argues that the widest minimum in the loss landscape is not the best in terms of generalization. The authors provide theoretical arguments and claim that there exists an optimal width beyond which generalization can be poor. Synthetic simulations are presented to support these claims.

The authors employ Fisher Information to characterize the optimal width or the curvature around the minimum. The fact that the determinant of the Fisher Information Matrix is invariant to parametrization, under certain conditions, serves as the motivation to design an objective Bayesian prior called Jeffrey's prior.

The motivation and the theoretical arguments are interesting, but the paper lacks in presentation and sufficient empirical evidence is also lacking to get fully convinced by the claims.

The authors should discuss the architecture design choices used for the synthetic data-generating model. Why are the last 3 layers of the larger model comprise of linear mappings?

Fig 1 is not clear. What does n=23 signify in the caption? More discussion is needed to describe "intersection of the likelihood values", "Difference in update Step" and "density is placed around 0" in section 5.




**Experience Assessment:**

I have read many papers in this area.

**Review Assessment: Checking Correctness Of Derivations And Theory:**

I assessed the sensibility of the derivations and theory.

**Review Assessment: Checking Correctness Of Experiments:**

I assessed the sensibility of the experiments.

**Review Assessment: Thoroughness In Paper Reading:**

I made a quick assessment of this paper.

---

> ### Author Response · Authors · 2019-11-13
> **Response to Official Blind Review #2**
>
> We thank the reviewer for their constructive feedback.
>
> Q1) The authors should discuss the architecture design choices used for the synthetic data-generating model.
>
> A1) In light of the requests of Reviewer #3 we have updated our experimental procedure to be more general and utilize larger networks with more variance in their design. Please see the general comment on our updated experimental procedure above, titled "General Comment on Updated Experimental Procedure", as this was also raised by another reviewer. We will be more explicit about our design choices in the updated version of the paper and agree that this requires more discussion. Our aim, however, with the generating model was to create a complicated function for the training network to model. As a result we began with non-linear sigmoidal layers to create a complex function. The linear layer in the output on the generating model was then used to obtain the scalar output for the regression task. In our updated experimental procedure we utilize more general and larger generating networks. We also discuss this aspect in the general comment on our updated experimental procedure.
>
> Q2) Why are the last 3 layers of the larger model comprise of linear mappings?
>
> A1) This was merely to over-parametrize the model. Naturally any consecutive linear layers can equally be compressed into a single layer, however, the addition of more linear layers does increase the expressive power of the model and aids in overfitting. The impact of this design decision on the loss landscape is evident in the work on alpha-scaling [1] in which it is shown that by placing more weight on one layer of linear parameters while proportionally decreasing the weight on the following linear layer it is possible to move to an area in the landscape with different width but without changing the model behaviour. This is a direct result of the linear layers being over-parametrized and when parameter weight is spread over more linear layers wider landscapes will occur. In line with our work, we believe the wider areas to overfit more and, thus, the inclusion of more linear layers will help enforce that the training network overfits the training data.
>
> Q3) Fig 1 is not clear. What does n=23 signify in the caption?
>
> A3) n represents the number of datapoints, separate trainings run, in generating the figure. We will include this in the caption.
>
> Q4) More discussion is needed to describe "intersection of the likelihood values", "Difference in update Step" and "density is placed around 0" in section 5.
>
> A4) Thank you for pointing this out. We will expand on these points in the paper. We have elaborated on these points in another general comment above: "General Comment on Previous Experimental Procedure". In summary, however, we use the phrase "intersection of the likelihood values" to express the point at which the training network has the same error as the true data generating network on the noisy training data. We believe this to be the point at which the Jeffreys Prior parametrization is found in the loss landscape. To test our assertion that the Jeffreys Prior parametrization provides the optimal test performance we observe the number of parameter updates between where the Jeffreys Prior parametrization is found and where the minimum test error is found. We referred to this as the "Difference in update step". We then plot a histogram and kernel-density estimation (KDE) of the difference in update step from repeated trainings. We found a significant portion of the KDE was situated around the difference in update step of $0$ and said that "the density is placed around 0".
>
> [1] Dinh, Laurent, et al. "Sharp minima can generalize for deep nets." Proceedings of the 34th International Conference on Machine Learning-Volume 70. JMLR. org, 2017.

---

### Official Review · AnonReviewer3 · 2019-10-23
**Official Blind Review #3**

**Rating:** 1

**Review:**

The paper conjectures that the so called Jeffreys prior over the parameters of a neural network is the prior leading to the best generalization performance. The authors test this conjecture on an artificial task using a small neural network, and investigate the sensitivity of the results to noise.

I like the general idea of the paper and appreciate the very detailed exposition placing it in the context of other works. In particular, I enjoyed the summary showing the sometimes conflicting evidence for better generalization in either broader or sharper minima, and how it relates to the Jeffreys prior.

However, as I understood the paper, the main claim in page 5 Equation 4 “Thus we conjecture that a correct prior for a model would be:” is an *assertion* that Jeffreys prior is the correct prior to use over the parameter space of neural networks. While it is a possibility, the amount of empirical evidence presented does not (at least to me) provide strong enough justification.

On page 7, you say “This model was a neural network composed of one, 5 neuron, hidden layer which utilized a sigmoid activation function in its hidden layer and a linear activation in its scalar output layer.“, describing your experiment. I don't think this experiment is sufficiently large to convince me.

Furthermore, in Figure 1 values outside the optical cluster at 0.0 appear nonetheless. I am not sure how to judge the amount of spread I see, and what effect they have on the performance of the network.

In general I would like to see experiments on datasets and with architectures that are at least somewhat close to what people use in practice (at least in terms of the size of the task and the capacity of the net). That would give me more confidence that your conjecture is true. While I appreciate your detailed theoretical exposition, I think the amount of empirical evidence you provide is insufficient to back the claims. Considering the explicit instruction to judge papers exceeding 8 pages with a higher standard, I believe that the lack of a greater amount of empirical evidence is a significant deficiency of your otherwise very interesting work.

I encourage you to expand this paper and resubmit to another venue -- I believe it has a great potential.


**Experience Assessment:**

I have read many papers in this area.

**Review Assessment: Checking Correctness Of Derivations And Theory:**

I assessed the sensibility of the derivations and theory.

**Review Assessment: Checking Correctness Of Experiments:**

I assessed the sensibility of the experiments.

**Review Assessment: Thoroughness In Paper Reading:**

I read the paper at least twice and used my best judgement in assessing the paper.

---

> ### Author Response · Authors · 2019-11-13
> **Response to Official Blind Review #3**
>
> We thank the reviewer for their constructive feedback, and kind words.
>
> Q1) In Figure 1 values outside the optical cluster at 0.0 appear nonetheless. I am not sure how to judge the amount of spread I see, and what effect they have on the performance of the network.
>
> A1) The topic of Figure 1 and our experimental procedure was raised also by another reviewer and, thus, we answered this in a general comment: "General Comment on Previous Experimental Procedure". Please see this above, however, in summary this is due to the fact that networks will learn both true signal from the data as well as noise simultaneously while training. This distorts the results of our experiments when a significant degree is modelling early in training.
>
> Q2) In general I would like to see experiments on datasets and with architectures that are at least somewhat close to what people use in practice (at least in terms of the size of the task and the capacity of the net).
>
> A2) We are updating our experimental procedure to be more general and include larger models. We have made a general comment describing the new procedure above, titled: "General Comment on Updated Experimental Procedure". Thank you for this suggestion. We would value any further feedback or suggestions you may have for the new procedure.

---

### Author Response · Authors · 2019-11-13
**General Comment on Previous Experimental Procedure (Part 1 of 2)**

We are grateful for the time taken by the reviewers in helping to improve the quality of this paper and our work. The most common concern raised was that our experimental procedure was not sufficiently exhaustive to provide a compelling case for our theoretical arguments. We agree with these comments and in some cases the concerns raised with our experimental procedure are necessary and we aim to clarify their necessity in this general comment. We do, however, acknowledge that a more general and realistic experimental procedure was required. We have, thus, updated our experimental procedure and discuss this in the second general comment below.

Figure 1:
The most common concerns appear to be as a result of Figure 1. Thus, we will provide a brief summary of this Figure and clarify some terminology, such as "the intersection of the likelihoods", which were raised by Reviewer #2. The purpose of Figure 1 is to reflect the number of training steps between where the network achieves its minimum test set error and where the Maximum A Posteriori parametrization for the network using the Jeffreys prior is found (for brevity we will call this the Jeffreys prior parametrization). From the discussion on Page 8, and in particular using Equation 10, we reflect that the Jeffreys prior parametrization will result in the likelihood of the training neural network generating the training data and the likelihood of the true data distribution generating the training data being equal. In other words the Jeffreys prior results in a model with the same training error or variance as the true data distribution.

There are, however, two means by which the training network may reduce its error and increase its likelihood. Firstly it can model the true signal in the data, and secondly it can model the noise in the data. In reality, while training, the model will learn both noise and signal simultaneously. Conceptually if the model were to learn pure signal only it would model the true distribution identically, achieve a test error of $0$ (as no noise was added to this data) and obtain a training error equal to that of the true distribution. The model would then proceed to reduce training error by modelling the only information remaining, namely the noise and we would then see an increase in the test error. Unfortunately a normal training procedure is not as separable and as a result the network will model noise before learning all of the true signal in the data. Depending on the relative quantity of noise to signal being modeled two cases will occur. Case 1: The quantity of noise learned will dominate the true signal learned. In this case we will see the test error reach its minimum prior to the likelihoods intersecting. Case 2: The degree of true signal being learned dominates the degree of noise being learned. In this case the test error will continue decreasing slowly and, since true signal remains to be learned when the likelihoods intersect, the test error will be minimized after the likelihoods intersect. In summary, the Jeffreys prior parametrization equates the model and true distribution likelihoods. The fact that modelling the noise also increases the model likelihood distorts the point where the minimum test error is found. In cases where high quantities of noise are learned early in the training we observe the points outside the optical cluster at 0.0 (as well as an observably higher test error) as observed by Reviewer #2. We believe the symmetry of the density estimation of Figure 1 to be the main highlighting factor reflecting the fact that equating the likelihoods and using the Jeffreys prior parametrization results in minimum test error. In essence the difficulty of separating noise from true signal in data is the problem of overfitting, and one we aim to address in future work. We do, however, believe a contribution of this work to be the novel theoretical placement of where the minimum test error can be obtained in the loss landscape.

---

> ### Author Response · Authors · 2019-11-13
> **General Comment on Previous Experimental Procedure (Part 2 of 2)**
>
> We are in the process of running experiments on considerably larger networks, as requested by both Reviewer #2 and Reviewer #3, the details of which have been posted in another general comment below. While our new results are consistent with those provided in the original work, it appears the problems highlighted above were as a result of the training model being under-parametrized to fully learn the true signal and the noise. Thus, a trade-off occurred. The deeper models appear to be giving better results. The use of synthetic data is, however, still necessary for our experimental procedure. As stated above, we need to determine the point where the model variance is equal to the true distribution variance on training data (point where the likelihoods are equal). In all real-world datasets such ground truth information is not obtainable. Thus, the synthetic data afforded us the ability to precisely determine where the intersection of the likelihoods occurred, as well as the point where a minimum was reach in the error on the very large test set.
>
> Figure 2 and Figure 3:
> These experiments relying on the calculation of the Hessian matrix, however, are only computationally feasible (at least by our hardware constraints) on the smaller network. This is due to the Hessian of the network being an exceptionally large matrix for even a small network. Naturally there are techniques which aim to mitigate the size of the Hessian. These are, however, approximations and we believe the trade-off of using a smaller network to calculate the full and precise Hessian for our results as being worth-while.

---

### Author Response · Authors · 2019-11-13
**General Comment on Updated Experimental Procedure**

We are grateful for the time taken by the reviewers in helping to improve the quality of this paper and our work. Further, we acknowledge that the experimental networks in the original submission were too small and as a result we are obtaining experimental results on larger network architectures for Figure 1, with more variety in depth, width and the activation function used. We will update Figure 1 with these new results in the updated version. The new procedure is as follows. We create a randomly generated True network with depth between $5$ and $15$ layers. The widths of the model layers are randomly sampled between $5$ and $100$ neurons. The layers are sorted in descending order of width (so we have no encoder-type layers). We then prepend the $100$ neuron input layer and append the $1$ neuron output layer. At the moment all layers except the last are sigmoidal. This is the model used to *generate* data. We then randomly initialize our training network. The number of layers in this network is randomly chosen from the range of $[True Network Size+5, 25]$ to ensure we obtain a sufficiently large network to overfit the data. The widths of this network's layers are sampled from the range of $[True Networks Smallest Layer, 100]$. This is again to ensure the model is over-parametrized. The True networks parameter values as well as the Training networks initial values are sampled uniformly from $[-1.0, 1.0]$ with a random positional bias added to the True network parameters in the range of $[-0.5, 0.5]$. This bias is to ensure the Training network starts with a significant degree of error. Finally, we utilize randomly sampled values between $[0.0, 1.0]$ as input to the models, with a training batch size of $50$ datapoints and a test batch size of $500$. This data is input to the True network and we obtain the corresponding data labels as output. Lastly we add Gaussian noise to the Training data only (while the Test data remains clean) with a mean of $0$ and variance of $0.2$. The Training network is then trained to model the True network using this data and we observe the points where their likelihoods are equal and where the test error is minimized.

---

### Decision · Program_Chairs · 2019-12-19

**Decision:**

Reject

**Comment:**

There has been significant discussion in the literature on the effect of the properties of the curvature of minima on generalization in deep learning.  This paper aims to shed some light on that discussion through the lens of theoretical analysis and the use of a Bayesian Jeffrey's prior.  It seems clear that the reviewers appreciated the work and found the analysis insightful.  However, a major issue cited by the reviewers is a lack of compelling empirical evidence that the claims of the paper are true.  The authors run experiments on very small networks and reviewers felt that the results of these experiments were unlikely to extrapolate to large scale modern models and problems.  One reviewer was concerned about the quality of the exposition in terms of the writing and language and care in terminology.  Unfortunately, this paper falls below the bar for acceptance, but it seems likely that stronger empirical results and a careful treatment of the writing would make this a much stronger paper for future submission.